# Evaluation of Preventive Role of Systemically Applied Erythropoietin after Tooth Extraction in a Bisphosphonate-Induced MRONJ Model

**DOI:** 10.3390/medicina59061059

**Published:** 2023-05-31

**Authors:** Gonca Duygu, Gül Merve Yalcin-Ülker, Murat Günbatan, Merva Soluk-Tekkesin, Ceyda Özcakir-Tomruk

**Affiliations:** 1Oral and Maxillofacial Surgery Department, Faculty of Dentistry, Tekirdag Namık Kemal University, Tekirdag 59030, Türkiye; 2Oral and Maxillofacial Surgery Department, Faculty of Dentistry, Istanbul Okan University, Istanbul 34947, Türkiye; 3Department of Tumour Pathology, Institute of Oncology, Istanbul University, Istanbul 34093, Türkiye; 4Oral and Maxillofacial Surgery Department, Faculty of Dentistry, Yeditepe University, Istanbul 34728, Türkiye

**Keywords:** bisphosphonate, erythropoietin, medication-related osteonecrosis of the jaw, tooth extraction

## Abstract

*Background and Objectives*: In this experimental study, the prophylactic effect of systemically administered erythropoietin (EPO) in medication-related osteonecrosis of the jaw (MRONJ) was evaluated. *Materials and Methods*: The osteonecrosis model was established using 36 Sprague Dawley rats. EPO was systemically applied before and/or after tooth extraction. Groups were formed based on the application time. All samples were evaluated histologically, histomorphometrically, and immunohistochemically. A statistically significant difference in new bone formation was observed between the groups (*p* < 0.001). *Results*: When new bone-formation rates were compared, no significant differences were observed between the control group and the EPO, ZA+PostEPO, and ZA+Pre-PostEPO groups (*p* = 1, 0.402, and 1, respectively); however, this rate was significantly lower in the ZA+PreEPO group (*p* = 0.021). No significant differences in new bone formation were observed between the ZA+PostEPO and ZA+PreEPO groups (*p* = 1); however, this rate was significantly higher in the ZA+Pre-PostEPO group (*p* = 0.009). The ZA+Pre-PostEPO group demonstrated significantly higher intensity level in VEGF protein expression than the other groups (*p* < 0.001). *Conclusions*: Administering EPO two weeks pre-extraction and continuing EPO treatment for three weeks post-extraction in ZA-treated rats optimized the inflammatory reaction, increased angiogenesis by inducing VEGF, and positively affected bone healing. Further studies are needed to determine the exact durations and doses.

## 1. Introduction

By inhibiting bone osteoclastic activity, bisphosphonate compounds reduce complications, such as hypercalcemia and pathological fractures, which are caused by various malignant tumors. They are used to improve quality of life and reduce pain in cancer patients with bone metastases. Bisphophosnates (alendronate, risedronate (orally) and zoledronic acid, ibandronate (parenterally)) are also preferred for the prevention of fractures associated with osteoporosis. They have been shown to significantly reduce vertebral and nonvertebral fractures for patients with osteoporosis. Intravenous bisphosphonate therapy (zoledronic acid and ibandronate) is effective for the correction of malignancy-related hypercalcemia, as well as in the treatment of metastatic osteolytic lesions due to tumors associated with breast, prostate, and lung cancers and multiple myeloma [1,2]. Initially, bisphosphonate-related osteonecrosis of the jaw (BRONJ) was considered a possible complication of nitrogen-containing bisphosphonate-group drugs, particularly when intravenously administered [3]. Subsequently, it was thought that BRONJ could be caused by antiresorptive drugs, such as denosumab and sunitinib, and antiangiogenic drugs, such as bevacizumab. The American Association of Oral and Maxillofacial Surgeons (AAOMS) termed this ‘medication-related osteonecrosis of the jaw’ (MRONJ) [4].

According to the AAOMS, MRONJ is characterized by the formation of exposed bone areas in the maxillofacial region. It can persist for eight weeks or more in patients who have used, or are using, antiresorptive therapy alone or with immune modulators or antiangiogenic medications and have not undergone radiotherapy to the head and neck region [5]. In the last statement published in AAOMS 2022, fusion proteins (aflibercept), mammalian target of rapamycin (mTOR) inhibitors (everolimus), radiopharmaceutical drugs (radium 223), selective estrogen-receptor modulators (raloxifene), and some immunosuppressive drugs, such as methotrexate and corticosteroids, were the drugs known to cause MRONJ [5]. 

Although it is not exactly known why bisphosphonates cause osteonecrosis, particularly in the jaw, previous MRONJ studies have discussed theories related to changes in bone remodeling, excessive suppression of bone resorption, inhibition of angiogenesis, micro-fracture of the bone due to continuous microtrauma, bacterial invasion through the periodontium, suppression of congenital or acquired immunity, vitamin-D deficiency, bisphosphonate-related toxicity in soft tissues, and the presence of inflammation or infection [5]. Experimental studies have shown that bisphosphonates and other antiangiogenic drugs exert an inhibitory effect on vascular endothelial growth factor (VEGF) [6,7,8]. 

Erythropoietin (EPO), a 34-kDa glycoprotein hormone that belongs to the hematopoietic class I cytokine superfamily, is a growth hormone that stimulates neovascularization [9]. It has been used in various anemia treatments [10]. EPO plays an important role in the proliferation and differentiation of hematopoietic stem cells and hematopoietic progenitor cells [11]. Furthermore, its ability to stimulate bone formation and tissue preservation by reducing ischemia and inflammatory responses with non-hematopoietic effects has been studied [12]. EPO, which stimulates the differentiation of mesenchymal stem cells (MSCs) into osteoblasts, influences the interaction between osteoblasts and osteoclasts to increase bone formation [13,14]. EPO also inhibits excessive inflammation by regulating proinflammatory cytokines, such as tumor necrosis factor-alpha [15,16]. Additionally, in an experimental model of mandibular-distraction osteogenesis, EPO was found to increase the number of blood vessels, fibers, and osteoblasts, which, in turn, accelerated the healing rate and facilitated new bone formation [17]. EPO has been shown to increase new bone formation in mice and humans by stimulating MSCs [11,14]. Recent experimental animal studies have stated that systemic EPO treatment increases angiogenesis via the VEGF pathway [18] and acts as a therapeutic agent that increases chondrogenesis and angiogenesis in bone healing and improves skeletal regeneration [19,20,21,22]. As of yet, no treatment method that minimizes the risk of osteonecrosis, a potential complication after any surgical intervention, has been accepted as the gold standard. 

Considering the etiopathogenesis of MRONJ and the impact of EPO on bone healing and angiogenesis, this experimental study investigated the prophylactic preventability of jaw osteonecrosis after tooth extraction with systemic EPO administration. An animal model was used to simulate the use of bisphosphonate drugs in humans. 

## 2. Materials and Methods

### 2.1. Animal Care and Procedures

This study was conducted on 36 12-week-old female Sprague Dawley rats with an average weight of 200–220 g at the Yeditepe University Experimental Animals Research Center (Istanbul, Türkiye). Approval was obtained from the Yeditepe University Experimental Animals Local Ethics Committee (protocol number 2020-838). The rats were fed continuously with no food restrictions and had unlimited water. The rats were randomly divided into six groups (*n* = 6). On the first study day, rats in Groups I (control group) and II (EPO group) received intraperitoneal (IP) injections of sterile saline (SS; 0.1 mg/mL). Rats in Groups III (zoledronic acid [ZA]), IV (ZA+PostEPO), V (ZA+PreEPO), and VI (ZA+Pre-PostEPO) received IP injections of ZA at 0.1 mg/kg (Zolenat^®^ 4 mg/5 mL Novartis, Istanbul, Türkiye) [23]. The IP injection protocol began on the first day and was conducted three times a week for eight weeks (Table 1). Groups V and VI received an additional daily IP injection of EPO (NeoRecormon 500 IU, Novartis, Basel, Switzerland) for an additional two weeks prior to tooth extraction [19]. Groups II, IV and VI received an additional daily IP injection of EPO (NeoRecormon 500 IU, Novartis, Basel, Switzerland), and their injection protocol ran for three weeks. All injections were made in the midline (linea alba) where the leg connected to the body at the femoral head. 

The week after the last injection, 80–100 mg/kg ketamine hydrochloride (Ketasol^®^, Richterpharma, Wels, Austria) and 10 mg/kg 2% xylazine hydrochloride (Rompun^®^, Bayer, Leverkusen, Germany) were administered intramuscularly under veterinary supervision as general anesthetics before tooth extraction. Bacitracin and neomycin-sulfate ointment (5 g thiocillin, Abdi İbrahim, Istanbul, Türkiye) were applied to the eyes of each rat to prevent ophthalmic complications during general anesthesia. Under general anesthesia, the right lower-first molars were extracted using surgical asepsis and antisepsis. Post-extraction bleeding was controlled via a sterile gas tamponade and additional procedures were not applied. For three weeks after the extraction, rats in the control, ZA, and ZA+PreEPO groups began receiving daily IP injections of SS (0.1 mg/mL), and rats in the EPO, ZA+PostEPO, and ZA+Pre-PostEPO groups received daily IP injections of EPO (NeoRecormon 500 IU, Novartis, Basel, Switzerland). All experimental animals were sacrificed eight weeks post-tooth extraction (16 weeks from the beginning of the study) via the decapitation method. The mandible of each rat was dissected and removed. For the histopathological and histomorphometric evaluations, the mandibles were placed in 10% phosphate-buffered neutral formaldehyde for two weeks for fixation.

### 2.2. Histopathological and Histomorphometric Examination

All samples were kept in a decalcification solution (50% formic acid and 20% sodium citrate). The solution was changed every week, and the decalcification process took one month. After this, defective areas were dissected. After the samples were decalcified, paraffin blocks were obtained. These were cut into 4 μm slices and stained with hematoxylin and eosin (H&E). The stained sections were examined by a researcher, who was blind to the treatment of each group, using an Olympus BX60 microscope connected to a computer with a color-video camera (Tokyo, Japan). The Olympus Image Analysis System 5 was used for all measurements in the histomorphometric analysis. Images were taken with the camera at different magnifications, transferred to the computer, and calibrated. The histopathological presence of inflammation, foreign body reactions, and necrotic tissues were assessed. Inflammation was scored based on its intensity: 0 (absent), 1 (mild), 2 (moderate), and 3 (severe). In the histomorphometric examination, all socket (total bone [TB]) and vital bone (VB) areas were measured. An equation was used to calculate the formation of new bone: (VB/TB) × 100.

### 2.3. Immunohistochemical Examination

For immunohistochemistry, the paraffin blocks were cut into approximately 5 µm-thick sections on charged slides. First, the sections were penetrated and dried overnight in an autoclave (56 °C). They were deparaffinized with xylene for 30 min and washed with 99% alcohol for 15 min, then 96% alcohol and distilled water. The Histostain-Plus Bulk Kit (Zymed 2nd Generation, LAB-SA Detection System, 85-9043) and VEGF (Biorbyt Ltd., Cambridgeshire, UK) as the primary antibody were used in the study. Slides were incubated for 60 min. The negative control sections treated with phosphate-buffered antibodies were confirmed to be unstained. The secondary antibody was reacted for 25 min. AEC (ScyTek Laboratories, Inc. 205 South 600 West Logan, UT 84321, USA) chromogen was used to visualize the reaction. Lastly, the sections were counterstained with Mayer’s hematoxylin, coverslipped, and evaluated by a light microscope. A semiquantitative score system was used to evaluate the immunostaining data: 0 (−): 0–10% staining immunopositivity 1 (+): 10–25% staining immunopositivity2 (++): 25–50% staining immunopositivity3 (+++): 50–70% staining immunopositivity4 (++++): >75% staining immunopositivity

### 2.4. Statistical Evaluation 

Data were analyzed using the Statistical Package for Social Sciences for Windows v.23 (IBM, Armonk, NY, USA). Data normality was assessed using the Shapiro–Wilk test. Pearson’s chi-squared test was used to compare the inflammation and necrosis scores by group; multiple comparisons were performed with a Bonferroni-corrected Z-test. The Kruskall–Wallis H test was used to compare the new bone formation rate, which was not normally distributed within groups; multiple comparisons were examined via Dunn’s test. All results are presented as mean ± standard deviation and median (minimum–maximum). Non-normally distributed immunohistochemical staining intensity scores were compared between groups using the Kruskal–Wallis test, and multiple comparisons were performed with Dunn’s test. Statistical significance was assumed when *p* < 0.05.

## 3. Results

### 3.1. Histopathological Analysis 

A statistically significant difference in inflammation severity was found between the groups (*p* = 0.008). Table 2 shows the severity across the groups. Although inflammation was not observed in 66.7%, 83.3%, and 33.3% of the samples from the control, EPO, and ZA+Pre-PostEPO groups, respectively, and varying degrees of inflammation were observed in all the ZA, ZA+PostEPO, and ZA+PreEPO group samples (Table 2). A statistically significant difference was found in the distributions of the necrosis status between the groups (*p* < 0.001). Although no necrotic tissue was observed in the control and EPO groups, it was observed in all other groups. In particular, necrotic tissue was observed in 83.3% of samples from the ZA+PostEPO and ZA+PreEPO groups and 16.7% of samples from the ZA+Pre-PostEPO group (Figure 1 and Figure 2).

### 3.2. Histomorphometric Analysis 

Table 3 shows the new bone formation results of each group. A statistically significant difference was observed in the new bone formation rate between the groups (*p* < 0.001). The mean new bone formation rate in the control, EPO, ZA, ZA+PreEPO, ZA+PostEPO and ZA+Pre-PostEPO groups were 48.77, 58.44, 19.74, 32.63, 24.78, 45.74, respectively. When the control and EPO groups were compared, no significant differences were observed between the EPO, ZA+PostEPO, and ZA+Pre-PostEPO groups (*p* = 1, 0.402, and 1, respectively) but a significantly lower rate of bone formation was observed in the ZA+PreEPO group (*p* < 0.001). When the control and ZA group were compared, a significantly lower rate of bone formation was observed in the ZA group (*p* < 0.001). When the ZA group and EPO groups were compared, no significant difference was observed between the ZA+PostEPO and ZA+PreEPO groups (*p* > 0.05). However, a significantly higher rate of new bone formation was observed in the ZA+Pre-PostEPO group compared to the ZA group (*p* = 0.009). The pairwise comparison of the EPO groups revealed no significant difference in the new bone formation rate (*p* > 0.05).

### 3.3. Immunohistochemical Analysis

Microscopic images of VEGF immunohistochemical staining are shown in Figure 3, and the intensity scores for VEGF protein in the extraction sockets are shown in Table 4 and Figure 4. VEGF protein levels differed significantly between groups (*p* < 0.001). The VEGF intensity scores in the control group did not differ significantly from the intensity scores in the groups; EPO (*p* = 0.906), ZA (*p* = 1), ZA+PostEPO (*p* = 0.378) and ZA+PreEPO (*p* = 0.07). However, there was a statitistically significant difference between the control group and the ZA+Pre-PostEPO group (*p* < 0.001). Furthermore, there were not any statistically significant differences between the ZA group and ZA+PostEPO and ZA+PreEPO (*p* > 0.05), but there was a statistically significant difference between the ZA+Pre-PostEPO group and the ZA group (*p* = 0.017). There was no statistically significant difference between the other groups (*p* > 0.05). 

## 4. Discussion

MRONJ is a multifactorial clinical condition frequently encountered in maxillofacial surgery; however, its etiopathogenesis is only partially understood. MRONJ is associated with several etiopathogenic factors: apoptosis in osteoclasts, inhibition of bone remodeling, inflammation, and infection, and exerting antiangiogenic effects by reducing VEGF levels in circulation (5). The disruption in angiogenesis may be an important factor in the formation of this condition. Studies have demonstrated that drugs, particularly bisphosphonates, exert an inhibitory effect on VEGF (6,7,8). Therefore, the promotion of angiogenesis may present a promising way to prevent and treat MRONJ both experimentally and clinically. 

Many studies have been conducted to develop another effective therapeutic modality. The use of some materials, including platelet concentrates [24], bone morphogenetic proteins (BMPs) [25], teriparatide [26] and cell therapy with MSCs [27] showed a positive effect on MRONJ. The preventive and curative treatment of MRONJ presents a significant challenge. The current study was designed to investigate whether EPO treatment had a prophylactic effect in preventing the formation of experimental MRONJ. EPO was selected due to its ability to modulate inflammation, thereby promoting osteoclastic bone activity, and increasing angiogenesis. It is thought that systemic EPO treatment could both prevent osteonecrosis in the jawbone and help treat existing osteonecrosis cases. Studies have shown that topically applied EPO may be a potential preventive and therapeutic agent for BRONJ treatment in humans [28]. However, although previous studies have evaluated the osteogenic and angiogenic effects of EPO in different bone-defect models, none have assessed the role of systemic EPO application in preventing MRONJ. 

EPO is a glycopeptide molecule that acts as a renal hormone. Its most well-known function is the regulation of erythropoiesis. However, in recent years, it has been redefined as a multifunctional molecule due to its regenerative and protective properties in various tissues, including in the nervous system, heart, intestine, kidney, skin, and retina [29,30]. Although the literature describes many pathological conditions in which EPO treatment has been used, the appropriate dose and treatment duration remain unclear. For example, while a single dose of EPO therapy may positively impact wound healing, repeated high doses (5000 U/kg/day) may not provide additional benefits to tissue protection and may even impair this function [31]. This is likely due to the impact of increased hematocrit on the rheological blood properties, nutrient perfusion, and metabolic functions of injured tissue. 

A study of a closed-femur fracture model in mice demonstrated that high-dose EPO (5000 U/kg) administration for five days improved early endochondral ossification and mechanical strength [32]. However, although high-dose short-term EPO administration was effective in the treatment of closed fractures, this effect disappeared after five weeks [32]. A study evaluating the effect of low-dose EPO treatment found that it positively affected the radiological, histological, and biomechanical healing of experimentally created segmental defects in mice [28]. This study also found that this effect on bone healing was not limited to the early period, unlike high-dose short-term use [28]. Therefore, long-term low-dose EPO treatment is preferable. No complications were observed during the experimental procedures in the current study. Unfortunately, it is impossible to determine the most effective dose from our study, which focused on the effects of the minimum effective dose injection on MRONJ. Further experimental studies are needed to determine the different effective EPO doses for MRONJ treatment.

Inflammation is essential to start the bone-healing response, and its modulation is necessary for the continuation of this process [33,34]. However, it is unknown whether the inflammatory response positively or negatively affects healing [35]. The inhibition of inflammatory cytokines following fractures and soft-tissue damage accelerates new bone formation and impacts the biomechanical properties of bone [36], while excessive inflammation is known to stimulate bone resorption and reduce bone formation [37]. For instance, following tooth extraction, neutrophils are immediately recruited and produce activated myeloperoxidase (MPO), an inflammatory marker, via phagocytosis [34]. Tissue homeostasis can be disrupted by the excessive or low accumulation of neutrophils [38]. In an experimental study of EPO-treated mice, decreased MPO immunostaining after tooth extraction has suggested that EPO may be involved in the modulation of early inflammation in several diseases [39,40,41]. Based on the data in the literature, and considering the inevitable role of inflammation in the formation and resolution of MRONJ, EPO may be effective in the treatment of this condition.

In the current study, the inflammation rates observed in the control, EPO, and ZA+Pre-PostEPO groups were significantly lower compared to the ZA, ZA+PostEPO, and ZA+PreEPO groups. Previous studies have reported that EPO’s positive impact on bone is due to its anti-inflammatory [42] and angiogenic [43] effects. Similar to the results of the current study, previous experimental animal studies have demonstrated extreme inflammatory reactions within necrotic tissues after tooth extraction in animals treated with ZA [44]. Therefore, the application of EPO pre- and post-extraction can optimize the inflammatory reaction in the MRONJ models created with ZA. 

The mechanisms of neovascularization, which is responsible for the formation of new blood vessels, and angiogenesis, are two important components of regenerative processes, including bone regeneration [45] It is argued that EPO influences both mechanisms and, therefore, may affect bone regeneration [46] In previous studies, EPO has been shown to increase the number of stem cells and progenitor cells in bone marrow and EPO causes the release of endothelial progenitor cells (EPCs), which have been shown to contribute to the bone-healing process [47]. Garcia et al. reported that the number of EPCs in peripheral blood was significantly increased in animals treated with EPO [19]. Moreover, Ateşok et al. [48]. and Matsumoto et al. [49]. reported that EPCs not only promote angiogenesis but also stimulate osteogenesis by differentiating into osteoblastic cells. A recent study showed that EPO accelerates vascular-tissue formation and has an important role in bone formation after tooth extraction. [50].

Recent studies have highlighted the importance of VEGF for bone formation and repair [51]. EPO stimulates bone formation and osteoblast differentiation by increasing VEGF expression and blood-vessel formation. In an experimental study, EPO was found to stimulate bone formation, angiogenesis, cell proliferation, and VEGF expression [52]. Another study demonstrated that EPO treatment accelerated burn-wound healing by stimulating angiogenesis and extracellular matrix maturation [53] and increasing VEGF and endothelial nitric-oxide-synthase (eNOS) expression [53]. During the mid-stage of fracture healing (days 7–14), VEGF and eNOS have been reported to contribute to endochondral ossification [54]. These findings support the view that EPO may promote endochondral ossification via nitric-oxide- (NO) and VEGF-dependent pathways [32]. Holstein et al. demonstrated radiologically and histologically that EPO treatment had no effect on callus size in fracture healing; however, histologically evaluated callus density was higher after two weeks due to increased mineralization, which was assessed radiologically [32]. Furthermore, the accelerating effect of EPO on bone repair was evident in the endochondral ossification process, and the authors’ stated that long-term treatment would not provide additional fracture-healing benefits [32]. In the current study, it was observed that VEGF-intensity scores were higher in EPO groups compared to the control and ZA groups. EPO, applied pre- and post-tooth extraction, demonstrated significantly higher VEGF intensity scores than the other groups. The higher VEGF intensity score in the ZA+Pre-PostEPO group compared to the control indicated that EPO can prevent ZA-induced osteonecrosis in the extraction socket by inducing VEGF. 

Runx2 and osteocalcin, which are secreted by mature osteoblasts, are known biochemical markers and key transcription factors associated with osteoblast differentiation [54]. In an experimental model of osteonecrosis of the femoral head, increased expression of Runx2 and osteocalcin was observed both four and eight weeks after EPO treatment [54]. These results suggest that the positive effect of EPO on bone formation may occur via Runx2-mediated osteogenesis. Shiozawa reported that EPO increases bone formation by differentiating MSCs into osteoblasts and increasing BMPs production in hematopoietic stem cells [13]. It was observed that the positive reaction of EPO to Runx2 and osteocalsin in the extraction socket facilitated collagen synthesis and mineralized tissue formation. This condition has suggested that EPO has a crucial role in bone formation by modulating osteoblast differentiation [50].

No necrotic tissue was observed in the control and EPO groups in the current study. However, varying rates of necrotic tissue were observed in all other group samples. In terms of new bone-formation rates, samples from the ZA+Pre-PostEPO group, where EPO was applied pre- and post-tooth extraction, demonstrated significantly higher rates than the other ZA groups. Moreover, the absence of any difference with the control group indicated that the application of EPO pre- and post-tooth extraction might exert protective effects against MRONJ. Although there was no difference between the two groups (control and ZA+Pre-PostEPO group) in terms of new bone formation, a difference in VEGF intensity scores supports the view that EPO can prevent ZA-induced osteonecrosis in the extraction socket by inducing VEGF. In addition, although VEGF expression was high in the ZA+PostEPO and ZA+PreEPO, there was no significant difference in VEGF expression and new bone formation, suggesting that it has no protective effect when EPO is applied before or after extraction. 

In the ZA+PostEPO and ZA+PreEPO groups, EPO had no effect on inflammation or other parameters. However, the current study supports the idea that long-term post-extraction EPO application may have a preventive and therapeutic effect when also applied pre-extraction. EPO’s prophylactic efficacy in the ZA+PreEPO and ZA+Pre-PostEPO groups and therapeutic efficacy in the ZA+PostEPO and ZA+Pre-PostEPO groups were evaluated. The absence of any difference between the ZA+PreEPO and ZA+PostEPO groups could be explained by the short-term effect of EPO on bones. The difference between the ZA+Pre-PostEPO group and the other ZA+EPO groups can be explained by the fact that EPO administered pre- and post-tooth extraction both modulates the inflammation caused by the extraction and exerts its effects afterward. An experimental study evaluating the treatment efficacy of EPO reported that EPO accelerated the early-healing rate of extraction sockets in mice with periodontitis [50]. These early features of the EPO after extraction can be defined as follows: it maintains the integrity of human periodontal ligament fibroblasts (hPDLF); improves the proliferation of osteoblasts in the extraction socket; and controls inflammation and bone formation through angiogenesis. These results are consistent with previous studies, which support the effectiveness of EPO on bone-fracture healing and angiogenesis [19,54].

## 5. Conclusions

This was the first experimental study to investigate the prophylactic effect of systemic EPO application on MRONJ. EPO was administered at different times across the groups, and its effects on bone healing in ZA-treated rats after eight weeks was evaluated. The application of EPO, either pre- or post-tooth extraction, did not contribute to bone healing in ZA-treated groups; however, EPO could act as a prophylactic and therapeutic agent for bone healing when applied pre- and post-extraction. Administering EPO two weeks pre-extraction, and continuing EPO treatment for three weeks post-extraction in the ZA-treated rats, optimized the inflammatory reaction, increased angiogenesis by inducing VEGF, and positively affected bone healing. Further studies are needed. One of the limitations of the current study is that systemically applied EPO may exert systemic effects that may have impacted the results of this study. Evaluating the direct effect of local EPO application on bone healing may yield more useful and objective results.

## Figures and Tables

**Figure 1 medicina-59-01059-f001:**
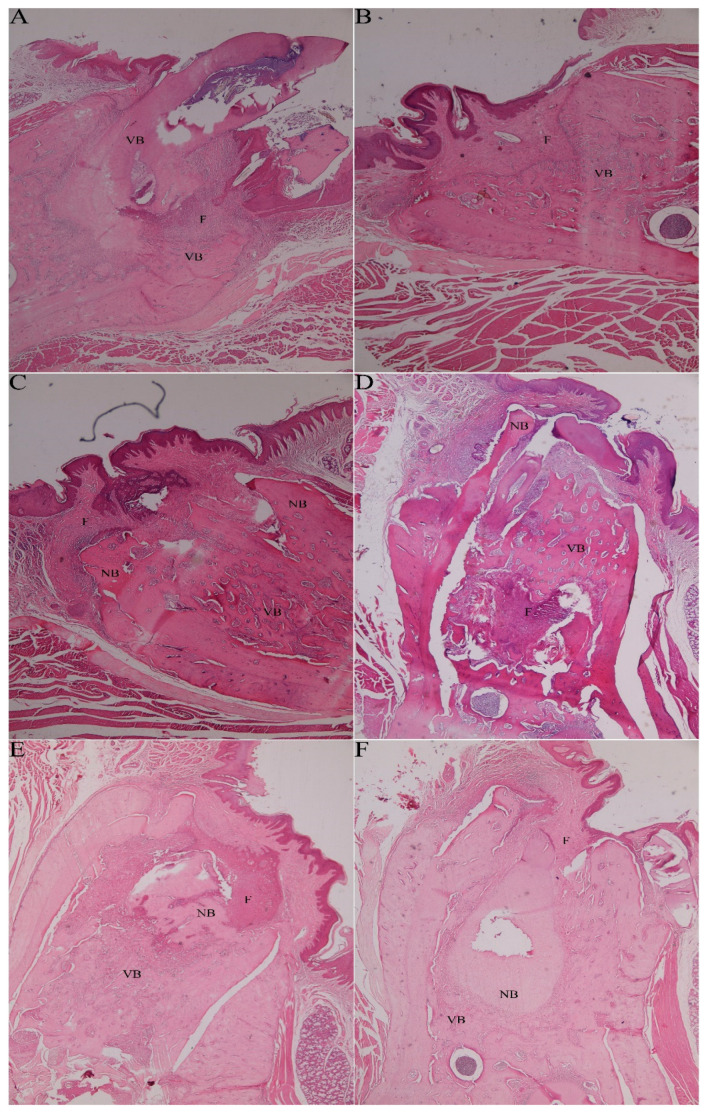
(**A**) Histological image of control groups showing vital bone and fibrosis in the extraction socket; (**B**) new vital bone formation and fibrosis filled the extraction socket of group EPO; (**C**) vital bone pieces surroundings by large necrotic bone and fibrosis were observed in the group of ZA; (**D**) various proportions of necrotic bone, vital bone and fibrosis were observed in the ZA+PostEPO group; (**E**) necrotic bone and fibrosis were surrounded by vital bone in the ZA+PreEPO group; and (**F**) various proportions of necrotic bone and fibrosis were surrounded by vital bone in the ZA+Pre-PostEPO group. Abbreviations in the images: VB: Vital Bone; NB: Necrotic Bone; F: Fibrosis (H and E, Original magnification ×40).

**Figure 2 medicina-59-01059-f002:**
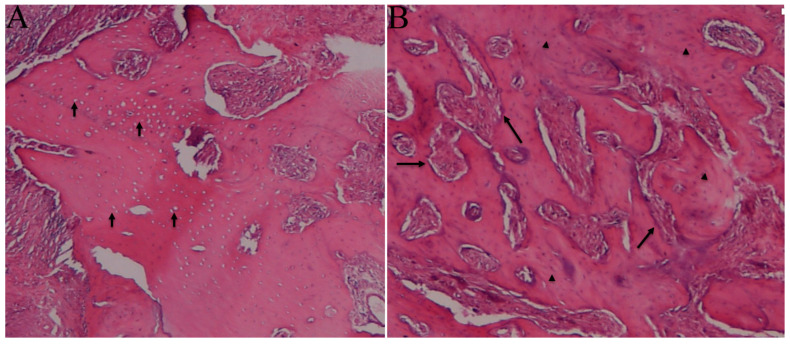
(**A**) Photomicrograph showing fragments of necrotic bone with empty lacunae (arrows); and (**B**) photomicrograph showing vital bone with osteocytes (arrowheads) and osteoblasts (arrow) around the trabeculae (H & E, Original magnification ×200).

**Figure 3 medicina-59-01059-f003:**
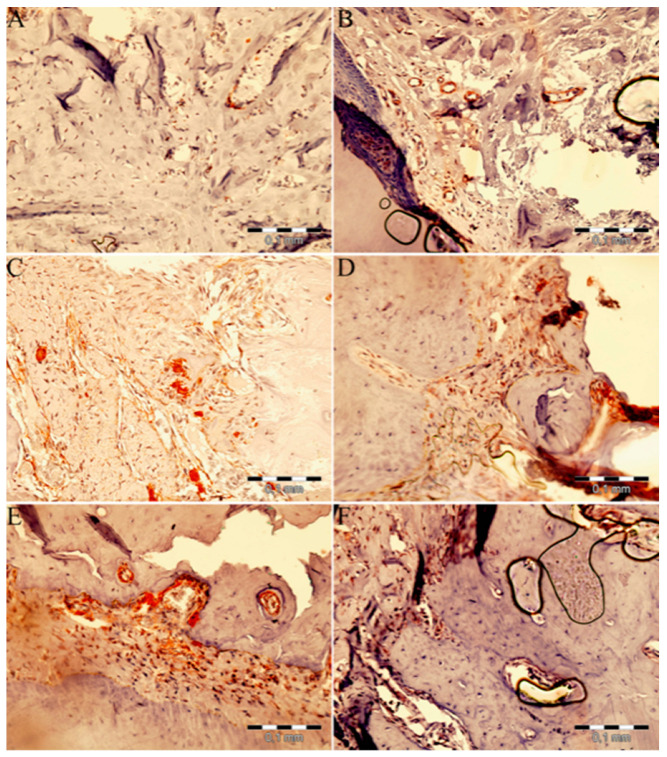
Demonstrative pictures of VEGF immunohistochemical staining:(**A**) control group; (**B**) EPO group; (**C**) ZA group; (**D**) ZA+PostEPO group; (**E**) ZA+PreEPO group; and (**F**) ZA+Pre-PostEPO group. More intense staining was observed in all the EPO groups compared to the control and ZA groups (VEGF, Original magnification ×400).

**Figure 4 medicina-59-01059-f004:**
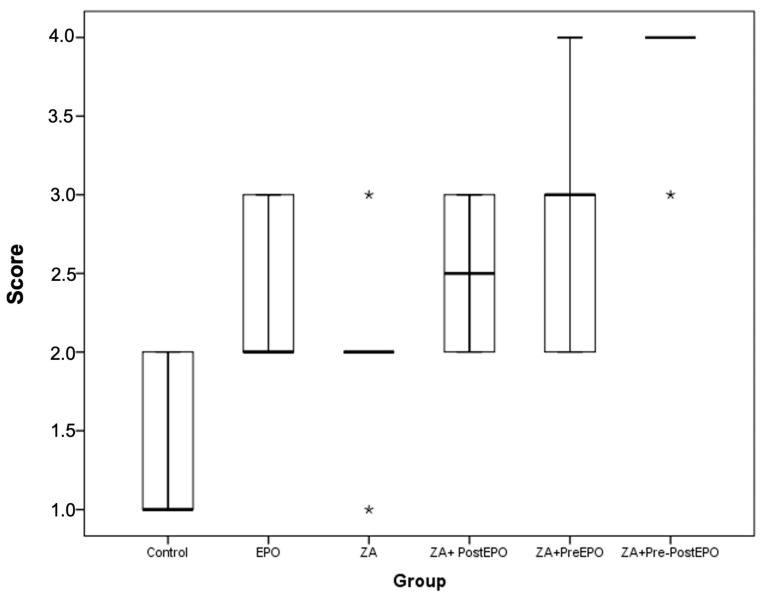
VEGF expression levels by groups; * *p* < 0.05.

**Table 1 medicina-59-01059-t001:** Procedures applied to rats in the control and experimental groups weekly. Abbreviations: SS: Sterile Saline; ZA: Zoledronic Acid; EPO: Erythropoietin IP: Intraperitoneal.

	SS	ZA	EPO
Weeks	(0.1 mg/kg 3 Times per Week IP)	(0.1 mg/kg 3 Times per Week IP)	(Daily 500 IU IP)
1	Groups I and II	Groups III, IV, V and VI	
2	Groups I and II	Groups III, IV, V and VI	
3	Groups I and II	Groups III, IV, V and VI	
4	Groups I and II	Groups III, IV, V and VI	
5	Groups I and II	Groups III, IV, V and VI	
6	Groups I and II	Groups III, IV, V and VI	
7	Groups I and II	Groups III, IV, V and VI	Groups V and VI
8	Groups I and II	Groups III, IV, V and VI	Groups V and VI
MANDIBULAR MOLAR EXTRACTION			
9	Groups I, III and V		Groups II, IV and VI
10	Groups I, III and V		Groups II, IV and VI
11	Groups I, III and V		Groups II, IV and VI
16	EUTHANASIA		

**Table 2 medicina-59-01059-t002:** Comparison of inflammation and necrosis by groups.

	Group 1 Control	Group 2 EPO	Group 3 ZA	Group 4 ZA+PostEPO	Group 5 ZA+PreEPO	Group 6 ZA+Pre-PostEPO	Test Statistics	*p* *
Inflammation								
0—Absent	4 (66.7)	5 (83.3)	0 (0)	0 (0)	0 (0)	2 (33.3)	31.295	0.008
1—Mild	2 (33.3)	1 (16.7)	3 (50)	4 (66.7)	2 (33.3)	4 (66.7)		
2—Moderate	0 (0)	0 (0)	1 (16.7)	2 (33.3)	3 (50)	0 (0)		
3—Severe	0 (0)	0 (0)	2 (33.3)	0 (0)	1 (16.7)	0 (0)		
Necrosis								
No	6 (100) ^a^	6 (100) ^a^	0 (0) ^b^	1 (16.7) ^ab^	1 (16.7) ^ab^	5 (83.3) ^ab^	25.969	<0.001
Yes	0 (0)	0 (0)	6 (100)	5 (83.3)	5 (83.3)	1 (16.7)		

* Pearson’s chi-squared test, frequency (percentage), ^a,b^: there is no difference between groups with the same letter.

**Table 3 medicina-59-01059-t003:** Comparison of the new bone-formation ratio (%) values of the extraction sockets.

	New Bone Formation Rate (%)	
	Mean ± SD	Median (Minimum–Maximum)
Control	50.06 (36.7–56.6) ^cd^	48.77± 6.87
EPO	59.15 (43.59–73.12) ^c^	58.44 ± 8.82
ZA	20.11 (16.74–22.22) ^b^	19.74 ± 2.08
ZA+PostEPO	33.08 (26.24–35.78) ^abd^	32.63 ± 2.75
ZA+PreEPO	22.6 (10.53–48.46) ^ab^	24.78 ± 12.11
ZA+Pre-PostEPO	45.58 (41.74–50.17) ^ac^	45.74 ± 2.9
Test Statistics	38.109	
*p*	<0.001	

F: One way ANOVA; Kruskal–Wallis Test ^a–d^: There is no difference between groups with the same letter, ±standard deviation.

**Table 4 medicina-59-01059-t004:** Comparison of the intensity levels of VEGF expression in the extraction sockets.

	Mean ± SD	Median (Minimum–Maximum)	Test Statistics *	*p*
Control	1.33 ± 0.52	1 (1–2) ^a^	22.853	<0.001
EPO	2.33 ± 0.52	2 (2–3) ^ab^		
ZA	2 ± 0.63	2 (1–3) ^a^		
ZA+PostEPO	2.5 ± 0.55	3 (2–3) ^ab^		
ZA+PreEPO	2.83 ± 0.75	3 (2–4) ^b^		
ZA+Pre-PostEPO	3.83 ± 0.41	4 (3–4) ^ab^		

* Kruskal–Wallis Test; ^a,b^: There is no difference between groups with the same letter.

## Data Availability

The datasets generated during and/or analyzed during the current study are available from the corresponding author on reasonable request.

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
