# Peer review of "Evaluation of Preventive Role of Systemically Applied Erythropoietin after Tooth Extraction in a Bisphosphonate-Induced MRONJ Model"

_medicina, 2023, doi:10.3390/medicina59061059_

Round 1

Reviewer 1 Report

To the authors:

I found your article to be well-written and interesting, and I believe that it holds significance for healthcare professionals. However, while the study included both histological and histomorphometric evaluations, additional experimental results from other evaluation methods may be required to fully support your hypothesis. Therefore, I recommend that you consider incorporating additional evaluation methods, such as imaging, biochemical, physiological, or molecular biology assessments, to further validate your claims. Nonetheless, your findings are promising, and I appreciate your contribution to the field.

Author Response

Response to Reviewer 1 Comments

Point 1: I found your article to be well-written and interesting, and I believe that it holds significance for healthcare professionals. However, while the study included both histological and histomorphometric evaluations, additional experimental results from other evaluation methods may be required to fully support your hypothesis. Therefore, I recommend that you consider incorporating additional evaluation methods, such as imaging, biochemical, physiological, or molecular biology assessments, to further validate your claims. Nonetheless, your findings are promising, and I appreciate your contribution to the field.

Response 1: Thank you for your suggession and positive critism. However, unfortunately, the budget of this project has been closed and we do not have extra budget for an additional evaluation method.

Reviewer 2 Report

The authors present an interesting article about the use of Erythropoietin in the prevention of MRONJ. The topic is fascinating and innovative, since the literature has been demonstrating that the pathophysiology of MRONJ may be related to vascular suppression caused by the use of these drugs. 

I suggest corrections below, and the realization of an IHC to answer the central question of the paper. 

Introduction:

L39: Please, specify which BP.

L43-47: Please improve MRONJ definition by using AAOMS 2022 position paper. It is not only using BP. It can be diagnosed in patients undergoing to antiresorptive therapy alone or in combination with immune modulators or antiangiogenic medications, without prior exposure to radiotherapy in the head and neck region.

M&M:

How did the authors decide about the size of sample? Can authors provide the power test?

Table 2. Could the authors provide the exact p value for each group?

Table 2. Necrosis. It is a little confuse when authors say:

1) There is no difference between groups with the same letter

2)  that group 6 is “ac”, when Group 4 and 5 are also “c”, and literally we can see the difference.

L168: Authors provide p <0.001 for table 3, and inside the table the value is different. Please explain.

Figure 1. Authors are encouraged to use higher magnification images in order to highlight histological differences between non-vital and vital bone.

# Also, use arrows to indicate some of these differences, such as irregularities on the surface of the bony trabeculae, absence of continuity of the cortical layer, bone fragment loosening, bony sequestrum (necrotic bone), and osteocytes with empty Haversian canals.

# Please arrange the images so that they were all in the same direction.

Also, please quantify the number of samples with epithelium and the ones without. The presence of epithelium is very important to demonstrate no bone exposure. Authors are encouraged to discuss this in the discussion section.

Authors are strongly encouraged to perform immunohistochemistry at least for PECAM-1 /CD34 (endothelial markers). Also, TRAP and OC are interesting too.

Author Response

Response to Reviewer 2 Comments

Point 1: Introduction: L39: Please, specify which BP. L43-47: Please improve MRONJ definition by using AAOMS 2022 position paper. It is not only using BP. It can be diagnosed in patients undergoing to antiresorptive therapy alone or in combination with immune modulators or antiangiogenic medications, without prior exposure to radiotherapy in the head and neck region.

Response 1: L39: BP type has been spesified as;

Bisphophosnates (alendronate, risedronate (orally) and zoledronic acid, ibandronate (parenterally)) also are prefered for the prevention of fractures associated with osteoporosis. It has been shown to significantly reduce in vertebral and nonvertebral fractures for patients with osteoporosis. Intravenous bisphosphonate therapy (zoledronic acid and ibandronate) is effective for the correction of malignancy-related hypercalcemia, as well as the treatment of metastatic osteolytic lesions due to tumors associated with breast, prostate, and lung cancers and multiple myeloma [1,2].

MRONJ defination has been improved and added.

According to the American Association of Oral and Maxillofacial Surgeons (AAOMS), medication-related osteonecrosis of the jaw (MRONJ) has been characterized by the formation of exposed bone areas in the maxillofacial region. It can persist for eight weeks or more in patients who have used or are using antiresorptive therapy alone or with immune modulators or antiangiogenic medications and have not undergone radiotherapy to the head and neck region [5].

Point 2: M&M: How did the authors decide about the size of sample? Can authors provide the power test?

Response 2: G-Power protocol was used in order to calculate the number of the rats (Reference: DOI: 10.1016/j.joms.2017.04.017).

Input:       Effect size f                                         =      1.2855618

                   α err prob                                           =      0.05

                   Power (1-β err prob)                       =      0.95

                   Number of groups                           =      6

Output:    Noncentrality parameter λ           =      39.6640594

                   Critical F                                             =      2.7728532

                   Numerator df                                    =      5

                   Denominator df                                =      18

                   Total sample size                             =      24

                   Actual power                                    =      0.9943477

At least 4 specimens in each group should be included with 95% confidence (1-α), 99.9% test power (1-β) and f=1,28556 effect size.

-How many rats were used in each group?

6 animals were included in each group in order not to risk the statistical evaluation due to the possibility of the animal loss during the experiments.

6 animals were included in each group in order not to risk the statistical evaluation due to the possibility of the animal loss during the experiments.

Point 3: Table 2. Could the authors provide the exact p value for each group? Table 2. Necrosis.

Response 3: Proportional values were used for histopathological evaluation. Pearson's chi-square test for the comparison of proportional values was deemed appropriate as mentioned in the material method section. In the SPSS program, when this program is used for proportional comparisons, a p-value is not obtained for pairwise comparisons. In the output of the program, letters indicates the groups that differ significantly from each other.

For histomorphometric analysis, p-values for each pairwise comparison have been added into the results section.

Point 4: L168: Authors provide p <0.001 for table 3, and inside the table the value is different. Please explain.

Response 4: Thank you for your attention. The error in the table 3 has been corrected.

Point 5: Figure 1. Authors are encouraged to use higher magnification images in order to highlight histological differences between non-vital and vital bone. # Also, use arrows to indicate some of these differences, such as irregularities on the surface of the bony trabeculae, absence of continuity of the cortical layer, bone fragment loosening, bony sequestrum (necrotic bone), and osteocytes with empty Haversian canals.

Response 5: Histological images with higher magnification have been added to the manuscript. Extra markings and specific images, which are showing the differences between healthy and necrotic/pathological tissues, have been added. 

Point 6: Please arrange the images so that they were all in the same direction. Also, please quantify the number of samples with epithelium and the ones without. The presence of epithelium is very important to demonstrate no bone exposure.

Response 6: Thank you for your additive criticism. The images have been arranged accordingly.

Point 7: Authors are encouraged to discuss this in the discussion section. Authors are strongly encouraged to perform immunohistochemistry at least for PECAM-1 /CD34 (endothelial markers). Also, TRAP and OC are interesting too.

Response 7: Thank you for your rightful suggession. However, unfortunately, the budget of this project has been closed and we do not have extra budget for an additional evaluation method.

Author Response

Response to Reviewer 3 Comments

Point 1:  First paragraph of the Introduction – It feels like it’s just blindly copied from the ‘AAOMS Medication-Related Osteonecrosis of the Jaw – 2022 Update Position paper’.

Response 1: Thank you for your suggestion. The introduction has been revised as your suggestion and highlighted in the text.

By inhibiting bone osteoclastic activity, bisphosphonate compounds reduce complications, such as hypercalcemia and pathological fractures, which are caused by various malignant tumors. They are used to improve quality of life and reduce pain in cancer patients with bone metastases. Bisphophosnates (alendronate, risedronate (orally) and zoledronic acid, ibandronate (parenterally)) also are prefered for the prevention of fractures associated with osteoporosis. It has been shown to significantly reduce in vertebral and nonvertebral fractures for patients with osteoporosis. Intravenous bisphosphonate therapy (zoledronic acid and ibandronate) is effective for the correction of malignancy-related hypercalcemia, as well as the treatment of metastatic osteolytic lesions due to tumors associated with breast, prostate, and lung cancers and multiple myeloma [1,2]. Initially, bisphosphonate related osteonecrosis of the jaw (BRONJ) was considered a possible complication of nitrogen-containing bisphosphonate group drugs, particularly when intravenously administered [3]. Subsequently, it was thought that BRONJ could be caused by antiresorptive drugs, such as denosumab and sunitinib, and antiangiogenic drugs, such as bevacizumab. AAOMS termed this ‘medication-related osteonecrosis of the jaw’ (MRONJ) [4].

Point 2: Line 104 -105: Groups V and VI received an additional daily IP injection of EPO (NeoRecormon 500 IU, Novartis, Switzerland), and their injection protocol ran for nine weeks – Table 1 shows EPO protocol for Group V to be 2 weeks and for Group VI to be 5 weeks peri operatively. Clarify. 

Response 2: The material and metods section (Line 104-105) has been corrected and has been highlighted.

Groups V and VI received an additional daily IP injection of EPO (NeoRecormon 500 IU, Novartis, Switzerland) for an additional two week prior to tooth extraction [19]. Groups II, IV and VI received an additional daily IP injection of EPO (NeoRecormon 500 IU, Novartis, Switzerland), and their injection protocol ran for three weeks.

Point 3:  Post-extraction bleeding was controlled via a sterile gas tamponade and left to resolve without additional procedures – Gas tamponade?

Response 3: Line 117: The sentence has been corrected.

Post-extraction bleeding was controlled via a sterile gas tamponade and additional procedures were not applied.

Point 4: Three weeks after the extraction, rats in the control, ZA, and ZA+PreEPO groups began receiving daily IP injections of SS (0.1 mg/ml), and rats in the EPO, ZA+PostEPO, and ZA+Pre-PostEPO groups received daily IP injections of EPO (NeoRecormon 500 IU, Novartis, Switzerland) – Started three weeks post extraction? Table 1 doesn’t correlate.

Response 4: The sentence at the line 117 has been corrected and the correlation between text and Table 1 has been provided. 

During three weeks after the extraction, rats in the control, ZA, and ZA+PreEPO groups began receiving daily IP injections of SS (0.1 mg/ml), and rats in the EPO, ZA+PostEPO, and ZA+Pre-PostEPO groups received daily IP injections of EPO (NeoRecormon 500 IU, Novartis, Switzerland).

Point 5: Table 2 – Percentage with decimal point is more suitable to read rather than using ‘comma’

Response 5: The statement has been altered accordingly.

Point 6:  As per the heading ‘Animal Care and Procedures’, the rats were divided into 6 groups (n=6) – So, every group will have 6 rats each? If so, why Table 2 (under Inflammation as well as Necrosis category) has total of 8 subjects per group (6x8 = 48)?

Response 6: Thank you for your attention. The error in the table 2 has been corrected according to text. An error occurred while copying the tables.

Point 7: How many rats had clinically exposed or probe-able bone (Criteria for established MRONJ) after 8 weeks?

Response 7: The osteonecrosis model defined in the literature was used (1). Also, necrosis was detected histologically and histomomorphometrically in ZA group. No clinical osteonecrosis was observed.

  1. Dayisoylu, E.H.; Üngör, C.; Tosun, E.; Ersöz, S.; Kadioglu Duman, M.; Taskesen, F.; Senel, F.Ç. Does an alkaline environment prevent the development of bisphosphonate-related osteonecrosis of the jaw? An experimental study in rats. Oral Surg. Oral Med. Oral Pathol. Oral Radiol. 2014, 117(3), 329-34.

Point 8: ‘Histomorphometric Analysis’ – Simplify. Same things are said repeatedly.

Response 8:Thank you for your suggestion. Histomorphometric Analysis section has been reorganised accordingly and highlighted.

Table 3 shows the new bone formation results of each group. A statistically significant difference was observed in the new bone formation rate between groups (p < 0.001). The mean new bone formation rate in the control, EPO, ZA, ZA+PreEPO, ZA+PostEPO and ZA+Pre-PostEPO groups were 48.77, 58.44, 19.74, 32.63, 24.78, 45.74, respectively. When the control and EPO groups were compared, no significant differences were observed between the EPO, ZA+PostEPO, and ZA+Pre-PostEPO groups (p = 1, 0.402, and 1, respectively) but a significantly lower rate of bone formation was observed in the ZA+PreEPO group (p < 0.001). When the control and ZA group were compared, a significantly lower rate of bone formation was observed in the ZA group (p < 0.001). When the ZA group and EPO groups were compared, no significant difference was observed between the ZA+PostEPO and ZA+PreEPO groups (p > 0.05). However, a significantly higher rate of new bone formation was observed in the ZA+Pre-PostEPO group compared to the ZA group (p = 0.009). The pairwise comparison of the EPO groups revealed no significant difference in the new bone formation rate (p > 0.05).

Point 9: No significant difference between ‘Control - EPO’ and ‘Control - ZA+Pre-PostEPO’. However, very strong significant p value of 0.009 between ‘EPO - ZA+Pre-PostEPO’, even with a sample size of 6 subjects per group means extremely replicable evident results. Are the stats justified?

Response 9: This strong significant p value of 0.009 between ZA-ZA+pre-postEPO, not EPO-ZA+Pre-PostEPO.

Point 10: Table 3 - Percentage with decimal point is more suitable to read rather than using ‘comma’

Response 10: Tablo 3  has been altered accordingly.

Point 11: For example, while a single dose of EPO therapy may positively impact wound healing, repeated high doses (5,000 U/kg/day) may not provide additional benefits to tissue protection and may even impair this function – Reference?

Relevant 11: Relevant reference has been added. And the references have been updated.

  1. Sorg, H.; Krueger, C.; Schulz, T.; Menger, MD.; Schmitz, F.; Vollmar B. Effects of erythropoietin in skin wound healing are dose related. FASEB J. 2009, 23(9):3049-58.

Point 12: Therefore, long-term low-dose EPO treatment is preferable. No complications were observed during the experimental procedures in the current study (Lines 224-225). So based on what protocol was the dosage fixed and EPO administered only for 3 weeks post extraction and not for the whole 8 weeks?

   Relevant 12: There is no study in the literature about systemically EPO injection for MRONJ treatment. There is not any protocol about this. However,   experimental designs in the literature have used a lower dose applied repeatedly for 5 to 7 days or for 2–10 weeks. There is still controversy regarding the appropriate dose and route of administration. In order to minimize animal stress from multiple injections, we aimed for the lowest possible dose and the shortest duration of treatment. Because of this, we adapted the protocol applied by Garcia et al. for fractura healing. (Garcia, P.; Speidel, V.; Scheuer, C.; Laschke. M.W.; Holstein, J.H.; Histing, T.; Pohlemann, T.; Menger, M.D. Low dose erythropoietin stimulates bone healing in mice. J. Orthop. Res. 2011, 29(2), 165-72.). They found that low dose (500 U EPO/kg/day) long-term EPO treatment significantly accelerates healing of small bone defects not only during the initial 2-week period but also during the later time course. We thought that systemically-injected EPO reaches target tissue in the optimal concentration to exert its positive effect on bone healing. Because of this the EPO injection was started 2 weeks before tooth extraction. The reason why it is only in the first 3 weeks post extraction is that Garcia et al. applied it for 2 weeks and 5 weeks in the long term and also the granulation tissue after tooth extraction is organized in the first 3 weeks of wound healing. We hypothesised that EPO accelerates these phase. Why not for the whole 8 weeks, because Garcia et al. state that long-term administration of EPO, even at low doses, increases hematocrit and may adversely affect tissue healing.

Point 13: More-over, AteÅŸok et al. [47]. and Matsumoto et al. [48]. reported that EPCs not promote angiogenesis but also stimulate osteogenesis by differentiating into osteoblastic cells.

Relevant 13: The sentence has been corrected and highlighted in the text.

Moreover, AteÅŸok et al. [47]. and Matsumoto et al. [48]. reported that EPCs not only promote angiogenesis but also stimulate osteogenesis by differentiating into osteoblastic cells.

Point 14: An experimental study evaluating the treatment efficacy of EPO reported that.. (Lines 311-312) – Reference?

                Relevant 14: Relevant reference has been added and highlighted.

An experimental study evaluating the treatment efficacy of EPO reported that EPO has accelerated the early healing rate of extraction socket in mice with periodontitis  [50].

Point 15: Systemic complications expected to arise in humans?

Relavant 15: Translation of th experimental studies about systemic EPO injection into clinical trials requires a physiological dosage of EPO in order to avoid complications, such as thrombo-embolism, cerebral convulsion/hypertensive encephalopathy and arterial hypertension. Repetitive EPO injections ranging from 500 to 5000 IU/kg were administered in different in vivo studies (1,2), which have a systemic effect and potential risk of adverse events. Therefore, testing of the effective and clinically safe dose of EPO is essential before clinical trials can be considered. Garcia et al. (2) evaluated daily systemic treatment of low dose (500 U/kg) EPO. They found that low dose EPO increased biomechanical stiffness and radiological density of the femoral osteotomy gap. They noted significantly greater haemoglobin concentrations in blood samples from EPO-treated animals without any observed side effect (2). In another study of Holstein et al. they applied a high dose of EPO (5000 U/kg) for a short time (6 days) and stated that treatment enhanced early endochondral ossification and mechanical strength in closed femoral fracture model in mice. 5000 U/kg EPO-treatment resulted in a slight, but not a significant elevation of the haemoglobin concentration after 6 days of treatment and no adverse effect was noted (1). The rather low dosage of 500U/kg was chosen in the present study in order to minimize the potential side effects of EPO with no side effects noted in the experimental group.

  1. Holstein, J.H.; Menger, M.D.; Scheuer, C.; Meier, C.; Culemann, U.; Wirbel, R.J.; Garcia, P.; Pohlemann, T. Erythropoietin (EPO): EPO-receptor signaling improves early endochondral ossification and mechanical strength in fracture healing. Life Sci. 2007, 13, 80(10), 893-900.
  2. Garcia, P.; Speidel, V.; Scheuer, C.; Laschke. M.W.; Holstein, J.H.; Histing, T.; Pohlemann, T.; Menger, M.D. Low dose erythropoietin stimulates bone healing in mice. J. Orthop. Res. 2011, 29(2), 165-72.

Point 16: Any effect/ comment on the soft tissue counterpart (Gingiva) as its poor vascularity leads to bone exposure.

Relevant 16: Microvascular mucosal abnormalities and soft tissue toxicity can be observed in MRONJ pathogenesis. However,  bone exposure was not seen in experimental animals. Unfornately, we focus more on osteonecrosis. Further studies are needed.

Point 17: However, long-term therapy would not have further benefitted fracture healing, as the EPO receptor was only observed during the early post-fracture time period. (Lines 313-315) – Does fracture healing and post extraction healing have the same mechanism?

Relevant 17: Thank you for your rightful criticism. Bone healing process is similar all of the body. However, fracture healing and post extraction healing process are different. According to the current literature, the mechanism of EPO on bone healing is similar in both physiological conditions. These phases are vascularization and inflammation. This section has been revised and highligted.

An experimental study evaluating the treatment efficacy of EPO reported that EPO has accelerated the early healing rate of extraction socket in mice with periodontitis (diÅŸ çekimi makalesi). These early features of the EPO after extraction can be defined as follows; It maintains the integrity of human periodontal ligament fibroblasts (hPDLF), improves the proliferation of osteoblasts in the extraction socket, and controls inflammation and bone formation through angiogenesis. These results are consistent with previous studies, which support the effectiveness of EPO on bone fracture healing and angiogenesis [19, 54].

Point 18: In multiple places throughout the article, EPO influencing fracture healing and endochondral ossification is mentioned. So how does EPO affect the healing of bone post extraction that happens via intramembranous ossification. As EPO mainly influences only endochondral ossification, does intramembranous ossification influenced in the same fashion?

Relevant 18: Recent study showed that EPO was shown to increase the early healing rate of the extraction socket in mice with ligature-induced periodontitis. EPO might involve in modulation of early inflammation, EPO induces and facilitates vascular tissue formation plays roles in bone formation, EPO has a modulating during the proliferation and differentiation of osteoblasts during healing and regeneration of bone. Bone healing mechanism is similar but not the same. Relevant literature has been added.

Point 19: The study explains the use of EPO in subjects undergoing extraction with bisphosphonate therapy. Maybe retitle the article to ‘Preventive Role of Systemically Applied Erythropoietin in Post extraction/ Dentoalveolar surgeries and Bisphosphonate-induced MRONJ’ for clarity?

Relevant 19: Thank you for the descriptive title and the title has been changed as your suggestion.

“Evaluation of Preventive role of systemically applied erythropoietin after tooth extraction in a bisphosphonate-induced MRONJ model”

Point 20: Any comments on the other MRONJ causing drugs and effects of EPO on such conditions?

Relevant 20: In the literature, I have not come across any studies with other MRONJ causing drugs. Most angiogenesis inhibitors, such as the monoclonal antibody bevacizumab and the kinase inhibitor sunitinib, which could also cause MRONJ, target the VEGF signaling pathway. Because EPO stimulates bone formation and osteoblast differentiation by increasing VEGF expression and blood vessel formation; it could be interprete that this drug might be also useful for preventing MRONJ caused by antiangiogenetic drugs. In an experimental study, EPO was found to stimulate bone formation, angiogenesis, cell proliferation, and VEGF expression [1]. Another study demonstrated that EPO treatment accelerated burn-wound healing by stimulating angiogenesis and extracellular matrix maturation [2]. and increasing VEGF and endothelial nitric oxide synthase (eNOS) expression [2].

  1. Xu, T.; Jin, H.; Lao, Y.; Wang, P.; Zhang, S.; Ruan, H.; Mao, Q.; Zhou, L.; Xiao, L.; Tong, P.; Wu, C. Administration of erythropoietin prevents bone loss in osteonecrosis of the femoral head in mice. Mol. Med. Rep. 2017, 16(6), 8755-8762.
  2. Galeano, M.; Altavilla, D.; Bitto, A.; Minutoli, L.; Calò, M.; Lo Cascio, P.; Polito, F.; Giugliano, G.; Squadrito, G.; Mioni, C.; Giuliani, D.; Venuti, F.S.; Squadrito, F. Recombinant human erythropoietin improves angiogenesis and wound healing in experimental burn wounds. Crit. Care. Med. 2006, 34(4), 1139-46.

Round 2

Reviewer 1 Report

Thank you for your response and the clear explanation of your budgetary constraints. I understand the challenges that such limitations can impose on a research project.

However, my concern remains that without additional evaluation methods, it is challenging to fully support your hypothesis. This is a significant issue as it goes to the heart of the robustness of your findings.

In light of this, I regret to inform you that I cannot recommend acceptance of your paper in its current form. The lack of additional evaluation methods leaves a significant gap in the methodology and weakens the overall strength of the paper.

Nevertheless, I still believe that your work has potential. I would encourage you to consider resubmitting your paper in the future once you are able to incorporate additional evaluation methods, whether by securing additional funding or finding other ways to carry out these important analyses.

I appreciate the hard work you've put into your research and your professional attitude in dealing with these issues. Thank you for your understanding.

Author Response

Authors are grateful for the contribution of reviewers; the manuscript was improved effectively owing to constructive criticisms of reviewers.  Thank you for your suggession and positive critism. We have used own budget for an additional immunohistochemical analysis. All changes in the text and the tables in the manuscript were highlighted.
I hope we meet the expectations as per your request.
Thank you for your time and interest.
Sincerely,

Gonca DUYGU, DDS, PhD
Tekirdag Namık Kemal University Faculty of Dentistry
Department of Oral and Maxillofacial Surgery
Tekirdag/Türkiye
E-mail gduygu@nku.edu.tr; duygu_gonca80@hotmail.com
Mobile: +905332427721

Reviewer 3 Report

Minor grammatical and spelling errors.

Rest looks fine.

Author Response

(The authors gave the same response as above.)
